# Comparative Metabolomics and Transcriptome Studies of Two Forms of *Rhododendron chrysanthum* Pall. under UV-B Stress

**DOI:** 10.3390/biology13040211

**Published:** 2024-03-24

**Authors:** Wang Yu, Fushuai Gong, Xiangru Zhou, Hongwei Xu, Jie Lyu, Xiaofu Zhou

**Affiliations:** 1Jilin Provincial Key Laboratory of Plant Resource Science and Green Production, Jilin Normal University, Siping 136000, China17604342645@163.com (X.Z.);; 2Faculty of Biological Science and Technology, Baotou Teachers’ College, Baotou 014030, China

**Keywords:** metabolome, transcriptome, UV-B stress, *R. chrysanthum*, glyceric acid, amino acids, carbohydrates

## Abstract

**Simple Summary:**

This experiment is based on the treatment of two forms of *R. chrysanthum* with UV-B radiation and transcriptomic and metabolomic analyses. It aims to explore how plants adapt to UV-B stress from metabolomic and transcriptomic perspectives. Through this comparative study, potential UV-B stress biomarker and UV-B-responsive metabolites were identified. The main concentration sites of some differentially expressed genes under UV-B stress were summarized. The metabolic synthesis pathway of *R. chrysanthum* under UV-B stress and the relationship network of key metabolite-related and synthetic enzymes were constructed. The experimental results show that domesticated *R. chrysanthum* has a stronger UV-B tolerance. By comparing the differences in response to UV-B stress between the two forms of *R. chrysanthum*, we provide a reference for breeding domesticated plants with UV-B tolerance characteristics. These results will help to study the complex mechanisms of plant adaptation to UV-B radiation.

**Abstract:**

*Rhododendron chrysanthum* Pall. (*R. chrysanthum*), a plant with UV-B resistance mechanisms that can adapt to alpine environments, has gained attention as an important plant resource with the ability to cope with UV-B stress. In this experiment, *R. chrysanthums* derived from the same origin were migrated to different culture environments (artificial climate chamber and intelligent artificial incubator) to obtain two forms of *R. chrysanthum*. After UV-B irradiation, 404 metabolites and 93,034 unigenes were detected. Twenty-six of these different metabolites were classified as UV-B-responsive metabolites. Glyceric acid is used as a potential UV-B stress biomarker. The domesticated *Rhododendron chrysanthum* Pall. had high amino acid and SOD contents. The study shows that the domesticated *Rhododendron chrysanthum* Pall. has significant UV-B resistance. The transcriptomics results show that the trends of DEGs after UV-B radiation were similar for both forms of *R. chrysanthum*: cellular process and metabolic process accounted for a higher proportion in biological processes, cellular anatomical entity accounted for the highest proportion in the cellular component, and catalytic activity and binding accounted for the highest proportion in the molecular function category. Through comparative study, the forms of metabolites resistant to UV-B stress in plants can be reflected, and UV-B radiation absorption complexes can be screened for application in future specific practices. Moreover, by comparing the differences in response to UV-B stress between the two forms of *R. chrysanthum*, references can be provided for cultivating domesticated plants with UV-B stress resistance characteristics. Research on the complex mechanism of plant adaptation to UV-B will be aided by these results.

## 1. Introduction

In recent years, due to the increase in greenhouse gas concentration, climate change is further aggravated, and the stratospheric ozone is affected accordingly, leading to an increase in solar ultraviolet radiation (especially UV-B, 280–350 nm) on Earth’s surface [1,2]. Although UV-B accounts for a small proportion of total UV, it is traditionally considered as a kind of stress because it may have various harmful effects on plants, such as reduced growth rate, partial inhibition of photosynthesis, and changes in plant biochemistry [3,4,5,6].

*Rhododendron chrysanthum* Pall. is an evergreen shrub that can withstand low temperatures of around −40 °C [7]. *R. chrysanthum* only grows in the highlands of Changbai Mountain, Jilin Province, China at an altitude of 1300 to 2650 m. The harsh climate and poor soil on the top of Changbai Mountain pose severe challenges for plants. *R. chrysanthum* has developed resistance to UV-B stress and other abiotic stress in the long process of evolution [8,9].

The large amounts of metabolites produced within plants play a great role in maintaining their own growth and reproduction and resisting stress. With the progress of science and technology, metabolomics has become an effective method to study plants, revealing how plants respond to stress at the metabolic level [10,11]. Plants can alter the effects of UV-B on themselves through their own internal metabolites and even other abiotic or biological interactions. For example, UV-B radiation promoted the accumulation of flavonoids in Okra, resulting in significant domestication changes in the reduction of its epidermal UV transmittance [12]. Five different photosynthetic active radiation (PAR) background colors determined the resistance mechanism of cucumber leaves to low-dose UV-B radiation [13]. Plants even increase their leaf mass per area due to the effects of UV-B radiation [14]. Numerous metabolomics studies have explored the ways in which different plants respond to UV-B stress, which affects the accumulation of many different metabolites in these plants [15,16,17]. There have been previous experiments involving the application of gas chromatography–time of flight mass spectrometry (GC–TOFMS) in the identification of the metabolome of *R. chrysanthum* after UV-B radiation treatment [18].

In fact, multiple genes are involved in controlling the tolerance of plants to ultraviolet radiation [19]. Transcriptomics is a means of studying genes and can comprehensively display information on gene structure and gene function, making it an important tool for identifying genes responsible for plant resistance to abiotic stresses [20,21]. Since *R. chrysanthum* does not have a genome that can be referenced, transcriptomics can be used to respond to gene expression and understand its expression following abiotic stress [22]. The union of metabolomics and transcriptomics can reveal, more comprehensively, the direct link between the regulation of gene expression and metabolic changes. By integrating the data of the two, multilevel information on biological phenomena can be obtained from the transcriptional level to the level of metabolites, which can help to reveal the molecular level interactions and the network regulatory mechanisms in organisms [23,24,25,26]. This experiment combines metabolomics and transcriptomics, which is important for a comprehensive understanding of the metabolic state of *R. chrysanthum* as well as clarifying the defense mechanism of *R. chrysanthum* in response to UV-B.

This study examined the differences between two forms of *R. chrysanthum* in response to UV-B. The purpose of this experiment is as follows: (I) identification of potential UV-B stress biomarker and UV-B-responsive metabolites in *R. chrysanthum*; (II) comparison of the metabolic responses of two forms of *R. chrysanthum* to UV-B radiation and their different tolerances; (III) to explore genes related to potential UV-B stress biomarker from a transcriptomic perspective and to propose a framework for their metabolic response under UV-B stress.

## 2. Materials and Methods

### 2.1. Plant Material and Treatment

*R. chrysanthum* belongs to the rhododendron family and grows in the highlands of Changbai Mountain (41°59′36″ N 128°04′39″ E). After transferring *R. chrysanthum* from its origin (Changbai Mountain) to the laboratory, explants were obtained by culturing stem segments of *R. chrysanthum* in 1/4 MS medium. These explants cultured in 1/4 MS medium were placed in the artificial climate chamber (18 °C (14 h light)/16 °C (10 h dark)) under the 50 µmol (photon) $m^{-2}$ $s^{-1}$ white light, with a relative humidity of 60%, and in the intelligent artificial incubator (25 °C (14 h light)/18 °C (10 h dark)), with a relative humidity of 60% under the white light. Seedlings grown in the intelligent artificial incubator were termed incubator seedlings (Figure 1A), and seedlings grown in the artificial climate chamber were termed domesticated plants (Figure 1B), with domestication conditions applied in reference to the previous experiments [18,27].

For PAR+UV-B treatment, 320 or 295 nm long-pass filters (Edmund, Filter Long 2IN SQ, Barrington, NJ, USA) were placed on the culture flask, respectively. For PAR treatment, the 400 nm long pass filter was placed on the culture flask. Corresponding to the transmittance function of the long pass filter, the actual UV-B and PAR irradiance of the sample were 2.3 W $m^{-2}$ UV-B and PAR of 50 µmol (photon) $m^{-2}$ $s^{-1}$. The artificial UV-B and PAR used in this study were derived from UV-B fluorescent tubes (Philips, Ultraviolet-B TL 20 W/01 RS, Amsterdam, The Netherlands) and white-fluorescent lamps (Philips, T5 × 14 W, Amsterdam, The Netherlands). Different long pass filters were used to obtain two modes of target radiation.

The two forms of seedlings were subjected to PAR treatment or PAR+UV-B treatment for 8 h a day. After two days of stress treatment, two forms of tested materials with similar growth were selected, and the leaves with intact edges, no damage, and no macula were placed in liquid nitrogen. For RNA-seq analysis of leaf samples preserved in liquid nitrogen, a mixed sampling strategy was used to eliminate inter-individual variation and the experiment was repeated three times. For GC–TOFMS analysis of leaf samples, the experiment was repeated six times. All plants were completely randomized for RNA-seq and GC–TOFMS analysis.

Control group M (A): PAR treatment for incubator seedlings cultured in intelligent artificial incubator. Experimental group N (B): additional UV-B radiation treatment for incubator seedlings cultured in intelligent artificial incubator. Experimental group Q (C): same additional UV-B radiation treatment for domesticated plants cultured in the artificial climate chamber. Both PAR and PAR+UV-B treatments lasted for two days, with 8 h of treatment per day (Figure 2).

### 2.2. Detection of Physiological Indexes of Experimental Radiation Treatment of R. chrysanthum

The biological replicates of six sets of plant samples under liquid nitrogen cryopreservation were used for experimental studies. Soluble sugar and amino acid contents were based on previous experiments and assayed via GC–TOFMS (Pegasus HT, Leco, St. Joseph, MA, USA), gas chromatography (7890B, Agilent, Santa Clara, CA, USA), and double-headed sample MPS2 (Gerstel, Muehlheim, Germany) [27]. The content of SOD was similarly assayed with reference to the experimental completion of the previous experiment [28,29].

### 2.3. Analyzing Metabolites Quantitatively and Qualitatively with GC–TOFMS

Metabolomics analysis was performed using the XploreMET platform (ver. 3.0, Metab—profile, Shanghai, China). As described above, samples were prepared in the same way and instruments were set up in the same way [30]. In general, each sample passed through the GC–TOFMS system (PEGA-sus HT, Leco, St. Joseph, MA, USA) with gas chromatograph (7890B, Agilent, Santa Clara, CA, USA) and a sample MPS2 with dual heads (Gerstel, Muehlheim, Germany) to complete the analysis.

The metabolomic data were compared with the JiaLib metabolite database using XploreMET (Metabo-Profile, version 3.0) software. This peak area represents the relative quantitative value obtained by integrating the chromatographic peaks of each metabolite.

### 2.4. Analysis of Metabolite Data

The PCA method was a multidimensional statistical analysis method used to recognize patterns while unsupervised. The filtered data were analyzed for principal component analysis (PCA) using R software (www.r-project.org (Email exchange on 13 February 2023)) [31,32]. MetaboAnalyst 5.0 software (https://www.metaboanalyst.ca/ (accessed on 27 July 2023)) was used to analyze metabolite differences in the sample via metabolic pathway enrichment analysis. The Pearson correlation coefficient (PCC) and K-means group analysis were performed using Metware Cloud (https://cloud.metware.cn (accessed on 6 May 2023)).

### 2.5. RNA-Seq Library Construction and Sequencing

Total RNA of the samples was extracted via the modified CTAB (Bio Basic, Toronto, ON, Canada) method using an Optimal Dual-mode mRNA Library Prep Kit (BGI, Shezhen, China). The appropriate amount of tissue was ground into powder under liquid nitrogen and transferred to 1.5 mL of preheated CTAB lysis reagent (add 2% β-mercaptoethanol) at 65 °C, and incubated at 65 °C for 15 min. After incubation, the tube was cooled to room temperature, then centrifuged at 4 °C with 12,000× *g* for 5 min.

The supernatant was transferred into a 2.0 mL EP tube. A total of 200 μL of chloroform/isoamyl alcohol (24:1) per mile littler of CTAB lysis buffer was added. The mixture was vortexed to mix and centrifuged at 4 °C with 12,000× *g* for 10 min.

Next, the clear upper aqueous layer was transferred to a 2.0 mL EP tube. An equal volume of phenol/chloroform/isoamyl alcohol (25:24:1) was added. The mixture was vortexed to mix and centrifuged at 4 °C with 12,000× *g* for 10 min. Then, the clear upper aqueous layer was transferred to a 2.0 mL EP tube. An equal volume of chloroform/isoamyl alcohol (24:1) was added. The mixture was vortexed to mix and centrifuged at 4 °C with 12,000× *g* for 10 min. The final clear upper aqueous layer was transferred to a new 1.5 mL EP tube and added with 2/3 the aqueous layer’s volume of isopropanol. The mixture was gently inverted to mix well and placed at −20 °C for 2 h. The RNA precipitation was collected through centrifugation at 4 °C with 12,000× *g* for 20 min. The supernatant was removed, and the precipitation was resuspended and washed with 1 mL of 75% ethanol. After washing, the precipitation was centrifuged again at 4 °C with 17,500× *g* for 3 min. The supernatant was discarded, and the precipitation was dried in the biosafety cabinet for 3–5 min. Finally, 20µL~200µL of DEPC-treated or RNase-free water was added to dissolve the RNA.

Total RNA was treated using a mRNA enrichment method [7]. The mRNA enrichment: mRNA with polyA tail was enriched with magnetic beads with OligodT and a Oligitex mRNA kit. rRNA removal: RNA was hybridized with a DNA probe, RNaseH was selectively digested the DNA/RNA hybrid strand, the DNA probe was digested with DNaseI, then the desired RNA was purified. The RNA obtained was fragmented by interrupting the buffer, random N6 primers were reverse transcribed, and the two-stranded cDNA was synthesized to form double-stranded DNA. The ends of the synthesized double-stranded DNA were flattened and phosphorylated at the 5’ end, forming a sticky end with an “A” at the 3’ end and connecting a bubbling junction with a protruding “T” at the 3’ end. The ligation products were PCR amplified by specific primers. The PCR product was thermally denatured into a single strand, and then a bridge primer was used to circulate the single-stranded DNA to obtain a single-stranded circular DNA library. AMPure XP beads were used to screen the cDNAs (~150 bp) before constructing the cDNA library. An RNA Library Prep Kit for Illumina1 (NEBNext1UltraTM) was used to create sequencing libraries (NEB, Ipswich, MA, USA). cDNA libraries were constructed using the Illumina HiSeq™ X-Ten platform (BGI, Shenzhen, China). The constructed libraries were inspected by Agilent 2100 Bioanalyzer and Abisteponeplus Real-Time CrAnalyzer, and sequenc ed after being qualified.

Nine cDNA libraries of *R. chrysanthum* in this experiment were produced by the Illumina HiSeq™ X-Ten platform based on the Shenzhen Huada Gene Science and Technology Research Co. (Shenzhen, China). The total raw reads of the nine samples ranged from 43.69 to 45.44 MB, and the Q20 was not less than 97.44%, indicating good sequencing quality. After data filtering, the total number of clean reads ranged from 42.21 to 43.51 MB. The Q20 values for the three comparison groups were 97.52–97.75%, 97.44–97.67%, and 97.65–97.73%. The Q30 values were 92.75–93.39%, 92.56–93.22%, and 93.09–93.36%. The results indicated that the sequencing results were of good quality and the data could be used for further assembly [27].

### 2.6. De Novo Assembly and Sequence Annotation

Data filtering: using BGI’s self-developed filtering software SOAPnuke (v1.4.0) [33], reads containing connectors (connector contamination), reads with an unknown base N content greater than 5%, and low-quality reads (defined as bases with a mass value less than 15 that account for more than 20% of the total base number of reads) were removed.

De novo assembly and quality assessment: The experimental reference genome version is Rhododendron_lapponicum. Since there is no reference genome sequence of *R. chrysanthum*, the filtered high-quality clean reads were de novo assembled (removing PCR repeats to improve assembly efficiency) using Trinity (v2.0.6) [34], and then used CD-HIT (4.6) to cluster and de-redundantly cluster the assembled transcripts to obtain Unigene. 

The quality of the assembled transcripts was assessed via BUSCO, which improved the results of the assembly assessment via comparison with conserved genes, indicating, to some extent, the completeness of the transcriptome assembly.

Gene Annotation: We annotated the assembled unigenes with seven major functional databases (KEGG, GO, NR, NT, SwissProt, Pfam, and KOG). Utilizing graphical representations, the KEGG database can depict various metabolic pathways and their interconnections. The KEGG Orthology (KO) system provided a cross-species annotation process by linking relevant information from molecular networks into the genome. By contrasting genes across various databases, we gain diverse insights into their structural and functional aspects.

### 2.7. Differential Expressed Genetic Analysis

Reference gene mapping: Clean reads were aligned to reference gene sequences using Bowtie2 (v2.2.5) [35], and RSEM (v1.2.8) [36] was used to calculate gene expression levels for individual samples. In order to identify the differences in gene expression in *R. chrysanthum* after UV-B radiation, the gene expression of each transcript was determined via the FPKM method. The DEseq2 method, based on the negative binomial distribution principle, was used to explore UV-B in *R. chrysanthum* differential genes for radiation [37]. This experiment was performed to detect differentially expressed genes according to the method described in the previously completed experiment [38]. In this study, Qvalue (adjusted *p* value) < 0.05, |lo$g_{2}$FC| ≥ 1 was used as a criterion for screening differentially expressed genes.

### 2.8. The Analysis of Statistical Data

IBM SPSS statistical software (http://www.ibm.com/analytics (accessed on 28 July 2023)) was used for statistical analyses. Letters indicate the significance level in the results. The data were analyzed for ANOVA, using 5% as the criterion for conducting Pearson’s correlation test, followed by one-way ANOVA.

## 3. Results

### 3.1. Metabolome Analysis of R. chrysanthum

The metabolites of *R. chrysanthum* treated with UV-B radiation were extracted. To investigate the metabolic response of chrysanthemum to UV-B stress, a non-targeted GC–TFOMS assay was used. The results show that 404 peaks were detected, of which 164 were known metabolites. These known metabolites were divided into 7 groups, including 40 amino acids, 37 carbohydrates, 38 organic acids, 10 nucleotides, 14 fatty acids, 10 lipids, and 15 other metabolites (Figure 3A).

PCA analysis reflected the characteristics of metabolomic multidimensional data through several principal components, among which the first principal component (PCA1) could explain 18.74% of the features of the original data set and the second principal component (PCA2) could explain 11.44% of the features of the original data set (Figure 3B). In addition, correlation analysis was conducted for 18 biological replicates, and Pearson’s correlation coefficient was used as the standard of biological replicate correlation. The heat map shows that the correlation of six biological replicates in group Q was weak, while that in group M and N was highly positive (Figure 3C). In conclusion, the experimental design and data were reliable and suitable for further analysis.

### 3.2. Detection of Differential Metabolites (DMs) in R. chrysanthum in the Presence of UV-B

Fold Change (FC) and *p* values were used to screen differential metabolites, and the metabolites meeting FC ≥ 1 or ≤0.67 and *p* < 0.1 were considered differential metabolites. The results show that there were 22 DMs in the N and M groups, among which 9 DMs were up-regulated and 13 DMs were down-regulated. In the comparison between group Q and group M, there were 38 DMs, among which 26 DMs were up-regulated and 12 DMs were down-regulated. In the comparison between group Q and group N, there were a total of 34 DMs, among which 23 DMs were up-regulated and 11 DMs were down-regulated (Figure 4A).

After removing the repeated DMs among the three groups, 54 DMs were obtained. Among the 54 DMs, 30% were (16/54) amino acids, 20% (11/54) carbohydrates, 18% (10/54) organic acids, 4% (2/54) nucleotide, 6% (3/54) fatty acids, 9% (5/54) lipids, and 13% (7/54) other metabolites (Figure 4B).

The clustering heat map results show that UV-B radiation increased the content of amino acids and organic acids in two forms of *R. chrysanthum*, and domesticated *R. chrysanthum* had more amino acids and organic acids with increased content after UV-B radiation. In addition, the fatty acid content of incubator *R. chrysanthum* increased after UV-B radiation; however, domesticated *R. chrysanthum* showed little change (Figure 4C).

### 3.3. Exploration of UV-B−Responsive Metabolites in R. chrysanthum

A Venn diagram was constructed to show the aggregation relationship of DMs in two forms of *R. chrysanthum* under UV-B stress (Figure 5A). Glyceric acid of both forms of *R. chrysanthum* showed a similar increasing trend after UV-B radiation, and, therefore, Glyceric acid was proposed as the potential UV-B stress biomarker. Removing the duplicated DMs between the control groups, a total of 26 DMs were obtained, of which nineteen DMs were consistently up-regulated after UV-B radiation and seven DMs were negatively feedback-regulated by UV-B radiation—these DMs were used as UV-B-responsive metabolites (Figure 5, Table 1).

Of these UV-B-responsive metabolites, most of the DMS were amino acids. Therefore, we further investigated the dynamics of these DMS under UV-B stress. UV-B treatment elevated most of the amino acid contents in both forms of *R. chrysanthum*, and some common amino acids that can play important roles, such as L-Tyrosine, L-Serine, N-Acetyl-L-aspartic acid, etc., showed similar response patterns under UV-B treatment—their contents all increased (Figure 6). It is also noteworthy that 3-Nitrotyrosine and L-Methionine showed diametrically opposite trends, decreasing in content after UV-B radiation. In contrast, organic acids all responded to UV-B stress in a positive response manner.

In summary, we identified a number of DMs with similar trends of changes in response to UV-B, with amino acids playing a more important role in the UV-B response.

### 3.4. Metabolic Pathways Enrichment Analysis

To investigate the different ways in which *R. chrysanthum* responds to UV-B stress, KEGG pathway enrichment analysis was performed on DMs from the three comparator groups. The was a total of 22 DMs (9 up-regulated and 13 down-regulated) from the NvsM comparator group, which were mainly enriched in Alanine, aspartate, and glutamate metabolism; Glyoxylate and dicarboxylate metabolism; Glycine, Serine, and threonine metabolism (Figure 7A). The 34 DMs (22 up-regulated and 11 down-regulated) produced by the QvsN comparator group were mainly enriched in Cysteine and methionine metabolism; starch and sucrose metabolism; and Neomycin, kanamycin, and gentamicin biosynthesis (Figure 7B). A total of 38 DMs (26 up-regulated and 12 down-regulated) from the QvsM comparison group were mainly enriched in Alanine, aspartate, Cysteine and methionine metabolism; Tyrosine metabolism; and other metabolic pathways (Figure 7C).

To comprehensively explore the metabolite metabolic network pathways of *R. chrysanthum* under UV-B radiation, KEGG-enriched pathways were analyzed in 54 DMs after the deletion of duplicate DMs from the three comparison groups. Overall, UV-B stress mainly affected amino acid metabolic pathways in the two forms of *R. chrysanthum*, such as Alanine, aspartate, and glutamate metabolism; Cysteine and methionine metabolism; Glycine, Serine, and threonine metabolism; and Tyrosine metabolism. Carbohydrate pathway metabolism, such as Pentose and glucuronate interconversions, and starch and sucrose metabolism were also affected to some extent (Table 2). Changes in these metabolic pathways are closely related to nitrogen and carbohydrate metabolism, suggesting that *R. chrysanthum* may respond to the stress effects of UV-B through alterations in carbon fixation pathways.

Finally, we constructed the metabolic reprogramming network of *R. chrysanthum* under UV-B stress. The results indicate that UV-B radiation promotes the production of the important intermediate carrier fumarate in the TCA cycle as well as increases the content of gamma-Aminobutyric acid, a positive response to abiotic stress, L-Asparagine, and N-Acetyl-L-aspartic acid as UV-B-responsive metabolites, all in a positive regulatory manner in response to UV-B stress (Figure 7D).

### 3.5. Transcriptomic Analysis of R. chrysanthum

In order to further explore the response mechanism of *R. chrysanthum* against UV-B stress at the molecular level, transcriptomic analysis was performed on two forms of *R. chrysanthums*. Analysis of the transcriptome sequencing data showed that the Q20 values of all samples involved in the experiment ranged from 97.44 to 97.75%, and the Q30 values ranged from 92.56 to 93.39% (Table 3), suggesting that the transcriptome data are plausible and suitable for downstream analysis. There was a total of 2348 differentially expressed genes (DEGs) in group A and group B, of which 1157 DEGs were revised up and 1191 DEGs were lowered. In the comparison between groups B and C, there were 1219 DEGs, of which 459 DEGs were up-regulated and 760 DEGs were down-regulated (Figure 8A).

The GO classification of DEGs to reflect the expression of DEGs: the trend of DEGs in the two comparison groups was similar, cellular process and metabolic process accounted for a higher proportion in biological processes, cellular anatomical entity accounted for the highest proportion in the cellular component, and catalytic activity and binding accounted for the highest proportion in the molecular function category (Figure 8B,C).

### 3.6. Transcriptomic Analysis of the Regulatory Network of Potential UV-B Stress Biomarker in R. chrysanthum

To explore the relationship between potential UV-B stress biomarker and DEGs under UV-B stress, a comprehensive network diagram was constructed to better reflect how potential UV-B stress biomarker and DEGs interact (Figure 9). Among a series of enzymes that regulate potential UV-B stress biomarker, aldehyde dehydrogenase involves an up-regulated DEG. (DL)-glycol-3-phase and glycol-3-phase O-acyltransferase each involve a down-regulated DEG. Diacylglycerol kinase involves two up-regulated DEGs. Acylglycerol lipase involves three down-regulated DEGs. There are six types of DEGs associated with 1-acyl-sn-glycerol-3-phase acyltransferase, including three up-regulated and down-regulated DEGs (Table 4).

### 3.7. Changes in Physiology of Domesticated R. chrysanthum

In this experiment, changes in soluble sugars and amino acids of two forms of *R. chrysanthum* were determined after UV-B radiation. The degree of alteration of soluble sugars and amino acids were measured separately and used to demonstrate their role in the domestication process. The results were as follows: after exposure to UV-B radiation, domesticated *R. chrysanthum* had a higher amino acid content than incubator *R. chrysanthum* but lower soluble sugars content than the latter. (Figure 10). The results indicate that domesticated *R. chrysanthum* is highly resistant to UV-B radiation.

Superoxide dismutase (SOD, EC 1.15.1.1) is an antioxidant enzyme that removes ROS and plays an important role in plant stress resistance. SOD has three types: Fe-SOD, Mn-SOD, and Cu/Zn-SOD. SOD expression levels in two forms of *R. chrysanthum* were determined in this experiment (Figure 11). The results show that the expression levels of the three SODs were significantly increased in domesticated *R. chrysanthum*, indicating that domesticated *R. chrysanthum* is remarkably resistant to UV-B radiation.

## 4. Discussion

Due to the degradation of the stratospheric ozone, the amount of UV-B radiation received by earth and plant surfaces has increased [39]. In addition, in terms of growth, development, and morphology, UV-B strongly affects plants [40,41,42]. In light of this, the effect of UV-B stress on *R. chrysanthum* deserves further study. In this experiment, domesticated *R. chrysanthum* was found to be more resistant to UV-B radiation [18]. In this study, it was observed that the biomarker of resistance to potential UV-B stress in *R. chrysanthum*’s UV-B response was Glyceric acid, and the UV-B-responsive metabolites affecting the UV-B response of *R. chrysanthum* were explored (Figure 5 and Figure 6).

Regarding the changes in the content of metabolites of both forms under UV-B stress, especially the dynamics of amino acids and carbohydrates, osmoregulation is a form of regulation when plants are under stress [43,44]. Under unfavorable conditions, to reduce the cytosol’s osmotic potential and prevent excessive water loss, solutes are actively accumulated in the cell. The accumulation of sugars plays a critical role in the adaptation of *R. chrysanthum* to UV-B stress, both from a growth and tolerance perspective. Glucose can act as a signal during plant stress [45,46,47]. Two different pathways are involved in amylose metabolism and sucrose metabolism in *R. chrysanthum* [48,49]. It is starch synthesis that takes place first in chloroplasts. As a result of the Calvin cycle, fructose 6-phosphate is converted into glucose 6-phosphate and starch follows. In the second pathway, the cytoplasm serves as a site for the synthesis and breakdown of sucrose [50]. The reduction in sugar transport and starch accumulation is likely to be related to the reduction in carbon flux for leaf sucrose synthesis [51]. The catabolism of starch can mitigate the effects of environmental factors on carbohydrates [52]. Therefore, *R. chrysanthum* can respond to UV-B through starch accumulation and catabolism [52]. Starch is one of the main nutritional components of *R. chrysanthum*. The growth, development, and physiological state of chrysanthemum are greatly affected by the changes in starch metabolites, which affect the quality of *R. chrysanthum*. In abiotic stress situations, glycolysis increases ATP production, which helps plants adapt to their environment [53]. When in an environment is unfavorable to the plant itself, the plant keeps the carbohydrate biosynthetic pathway from being disturbed through the TCA cycle [54]. Thus, the acquisition of greater UV-B resistance in domesticated *R. chrysanthum* may result from the activation of sucrose and starch metabolism.

Amino acids are a vital metabolite in plants, capable of functioning as intermediates in the metabolic process [55,56,57]. In previous studies, plants treated with UV-B radiation showed a significant increase in the expression of many amino acids internally [27]. In the present experiment, it was also found that most of the amino acid contents increased due to UV-B radiation, indicating that *R. chrysanthum* accumulated amino acids (Figure 4C). The experimental results show that most of the identified UV-B-responsive metabolites were amino acids (Table 1). In this study, prolonging the stress time significantly increased the Serine content in the experimental material. Serine is essential for photorespiration, and the accumulation of Serine suggests that *R. chrysanthum* can enhance photorespiration under UV-B stress [58]. Succinic acid, as an important intermediate in the TCA cycle, showed higher accumulation under UV-B stress, indicating that UV-B stress promoted the TCA cycle.

Moreover, the simplified metabolic model showed that plants may indirectly promote the expression of fumarate, an important intermediate of the TCA cycle, through the high expression of N-Acetyl-L-aspartic acid and L-Asparaginede (Figure 7D). Thus, the whole TCA cycle is promoted and the normal process of carbohydrate biosynthesis is guaranteed. This phenomenon is also consistent with the result that domesticated plants in this experiment have higher amino acid contents than incubator plants (Figure 10).

Superoxide dismutase (EC 1.15.1.1; SOD) catalyzes the dismutation of superoxide radicals to hydrogen peroxide and oxygen. In higher plants, SOD plays a major role in combating oxygen-radical-mediated toxicity. The sources of superoxide radical generation may be natural—that is, by-products of metabolic activities, including the electron-transport chain, or those induced by external agents including the ozone, UV-B, gamma rays, light-induced photoinhibitory conditions, or chemicals like paraquat or methyl viologen. Three classes of SOD have been identified, depending on the metals present at the active site: copper/zinc SOD, manganese SOD, and iron SOD. In higher plants, the Fe SOD has been isolated from plastids; Mn SOD from the mitochondrial matrix; and Cu/Zn SOD from the cytosol [59]. Under extreme conditions, plants are able to withstand harsh stresses by increasing the activity of antioxidant enzymes [60]. The results of this experiment show that domesticated *R. chrysanthum* had higher levels of the three SODs, indicating that this form of *R. chrysanthum* has a more positive response to UV-B as well as higher stress tolerance (Figure 11).

Glyceric acid (GA), i.e., 2,3-dihydroxypropanoic acid, is a simple but attractive derivative of glycerol because of its structure with two kinds of functional groups and chiral isomers. Moreover, GA is the oxidized product of glycerol, and its D-isomer is obtained as a phytochemical from tobacco leaves and some plant fruits [61]. Some biological effects of D-GA on ethanol metabolism have been reported. Ethanol metabolism in rat liver was accelerated by D-GA calcium salt [62]. Another aspect of GA is the protection of biological macromolecules. Radical scission of DNA was protected in the presence of 200 mM GA sodium salt [63]. In this study, Glyceric acid was defined as a UV-B-responsive metabolite via a comparative metabolomics analysis. Previous studies have shown that its transporter NPF8.4 is responsible for the isolation of photorespiratory carbon intermediate glycerol into vacuoles during nitrogen depletion, which elucidates a new function of photorespiration in nitrogen fusion [64]. The application of Glyceric acid in UV-B response may be related to photorespiration and deserves further study. Also, in this study, several enzymes, MGLL, GPAT1_2, GPP, DGK, plsC, and ALDII, were found to be closely related to Glyceric acid synthesis; however, the changes in these enzymes showed different trends (Figure 9). The inter-relationships between these enzymes related to Glyceric acid synthesis and their effects on Glyceric acid deserve further investigation.

This experiment analyzed the expression of DEMs in *R. Chrysanthum* under UV-B stress using transcriptome analysis. According to GO analysis, *R. Chrysanthum* has been severely affected by UV-B interference in terms of cellular atomicity, catalicacbactivity, binding, and cellular process. These results indicate that plants may repair the damage caused by ultraviolet radiation to plant nuclear organelles through their own defense mechanisms [65].

## 5. Conclusions

In this study, *R. chrysanthum* was used as the study subject for metabolomic analysis to study plant responses to UV-B caused by an ozone hole. In order to clearly respond to the mechanisms of plant responses to UV-B stress, the pathway map was constructed in order to show the interactions between the DMs. Potential UV-B stress biomarker and UV-B-responsive metabolites were screened by comparing DMs between groups. After UV-B irradiation, 404 metabolites and 93,034 unigenes were detected. A total of 26 of these different metabolites were classified as UV-B-responsive metabolites. Glyceric acid is used as a potential UV-B stress biomarker. The domesticated *Rhododendron chrysanthum* Pall. had high amino acid and SOD content. Enhanced UV-B resistance in tamed *R. chrysanthum* could stem from the stimulation of sucrose and starch metabolism. The study shows that the domesticated *Rhododendron chrysanthum* Pall. showed significant UV-B resistance. Transcriptomics results showed that both *R. chrysanthum* showed similar trends after UV-B stress, and both DEGs were mainly focused on cellular process, cellular anatomical entity, catalytic activity, and binding. And, it was also found that the enzymes MGLL, GPAT1_2, GPP, DGK, plsC, and ALDII closely affect the changes in the content of Glyceric acid in *R. chrysanthum*.

The results of this study further reveal the regulatory mechanism of *R. chrysanthum*’s resistance to UV-B radiation from a metabolomic perspective. In future studies, we will explore the molecular mechanisms of Glyceric acid and 26 UV-B-responsive metabolites and whether they play a role in the resistance of other plants to UV-B stress. The experimental results provide reference for the practical application of extracting a UV-B-resistant complex from plants.

## Figures and Tables

**Figure 1 biology-13-00211-f001:**
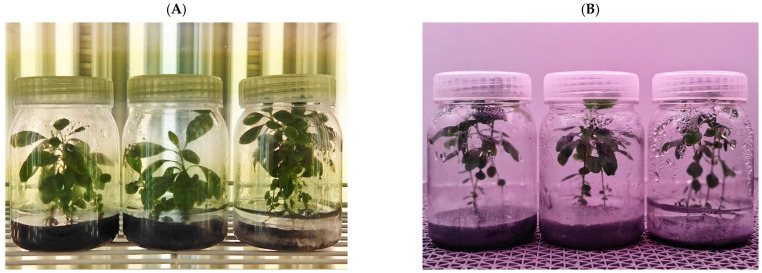
(**A**) The *R. chrysanthum* in the intelligent artificial incubator. (**B**) The *R. chrysanthum* in the artificial climate chamber.

**Figure 2 biology-13-00211-f002:**
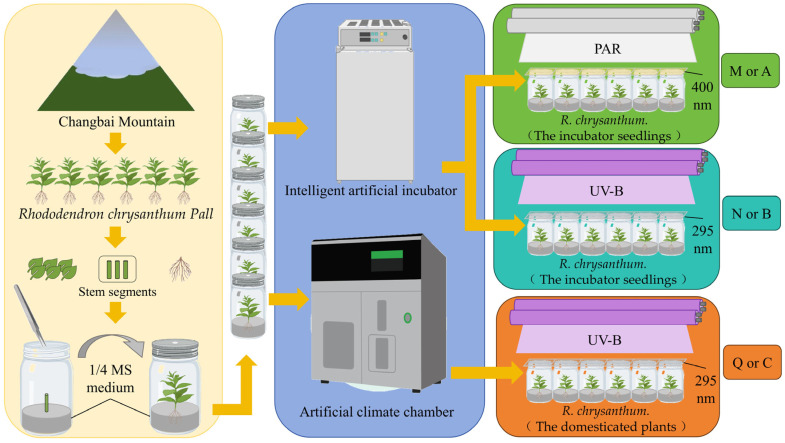
Experimental flowchart of *R. chrysanthum*. For metabolomics results, the three comparison groups are named “M, N, Q”; for transcriptomics results, the three comparison groups are named “A, B, C”.

**Figure 3 biology-13-00211-f003:**
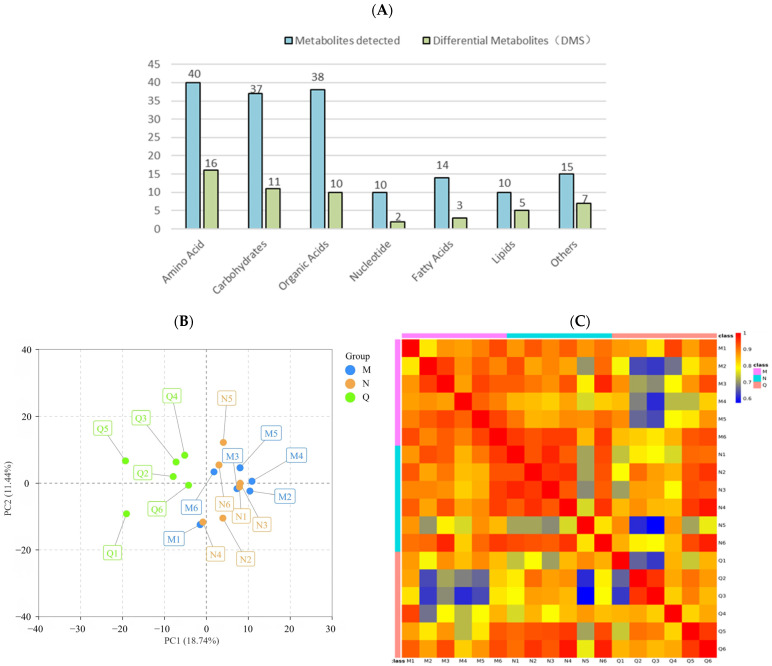
(**A**) The amount of metabolites and differential metabolites detected in different comparison groups. By using the hypergeometric test, the statistical significance was determined with *p* < 0.1. (**B**) Principal component analysis (PCA) of different comparison groups. (**C**) Heatmap of Pearson’s correlation coefficient. Different colors are used to represent different groups.

**Figure 4 biology-13-00211-f004:**
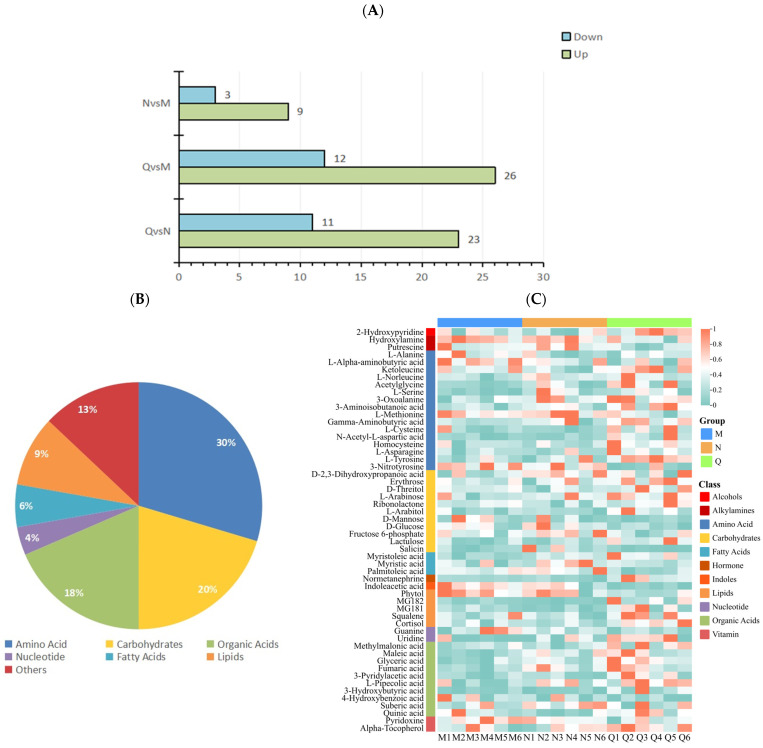
(**A**) Statistics of up-regulated and down-regulated DMs. (**B**) DM classification statistics. (**C**) Heatmap of clustering of 54 DMs. The accumulation level of each metabolite is reflected by the color of each square in the heat map. The more red the color, the higher the level of metabolite accumulation; the more green the color, the lower the level of metabolite accumulation.

**Figure 5 biology-13-00211-f005:**
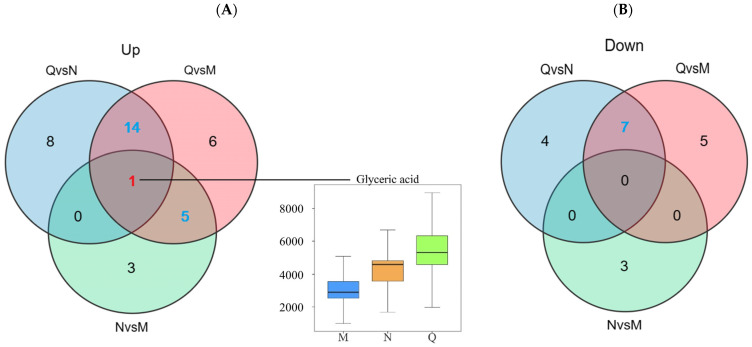
(**A**) Up-regulated DMs statistics. (**B**) Down-regulated DMs statistics. The red numbers in the venn diagram represent potential UV-B stress biomarker and the blue numbers represent UV-B-responsive metabolites. The top edge line of the boxplot represents the maximum value, the bottom edge line represents the minimum value, the top edge of the box is the top quartile, the bottom edges of the box are the bottom quartile, and the middle line of the box represents the median.

**Figure 6 biology-13-00211-f006:**
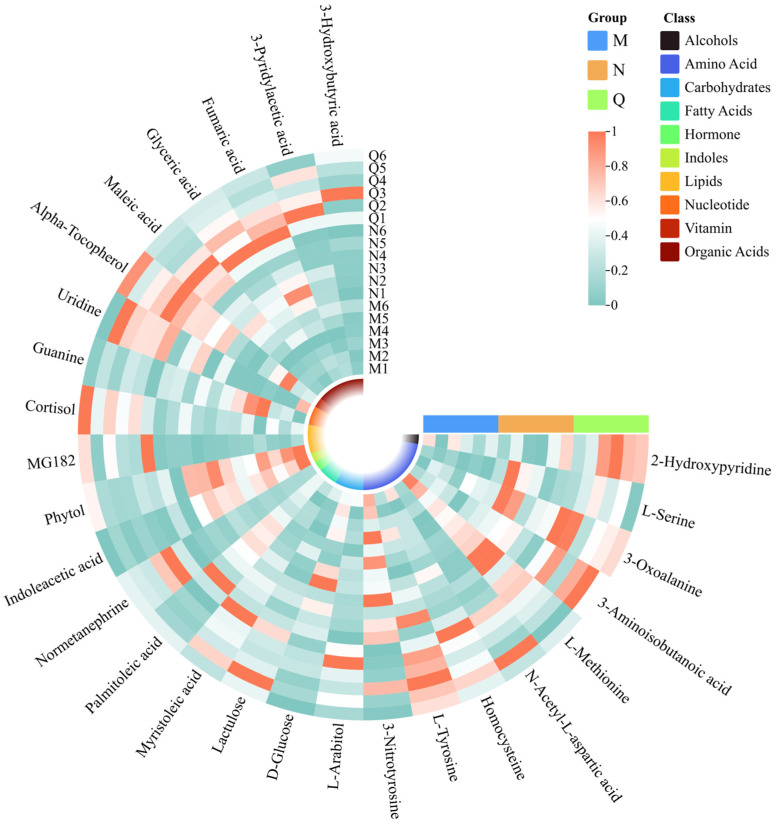
Clustering heatmap of potential UV-B stress biomarker and UV-B-responsive metabolites. The squares in the clustering heatmap are colored more towards red to indicate a higher content of the corresponding substance and more towards green to indicate a lower content. The ring in the center represents the subordinate category of the substance. The horizontal three-color scale represents the sample grouping categories.

**Figure 7 biology-13-00211-f007:**
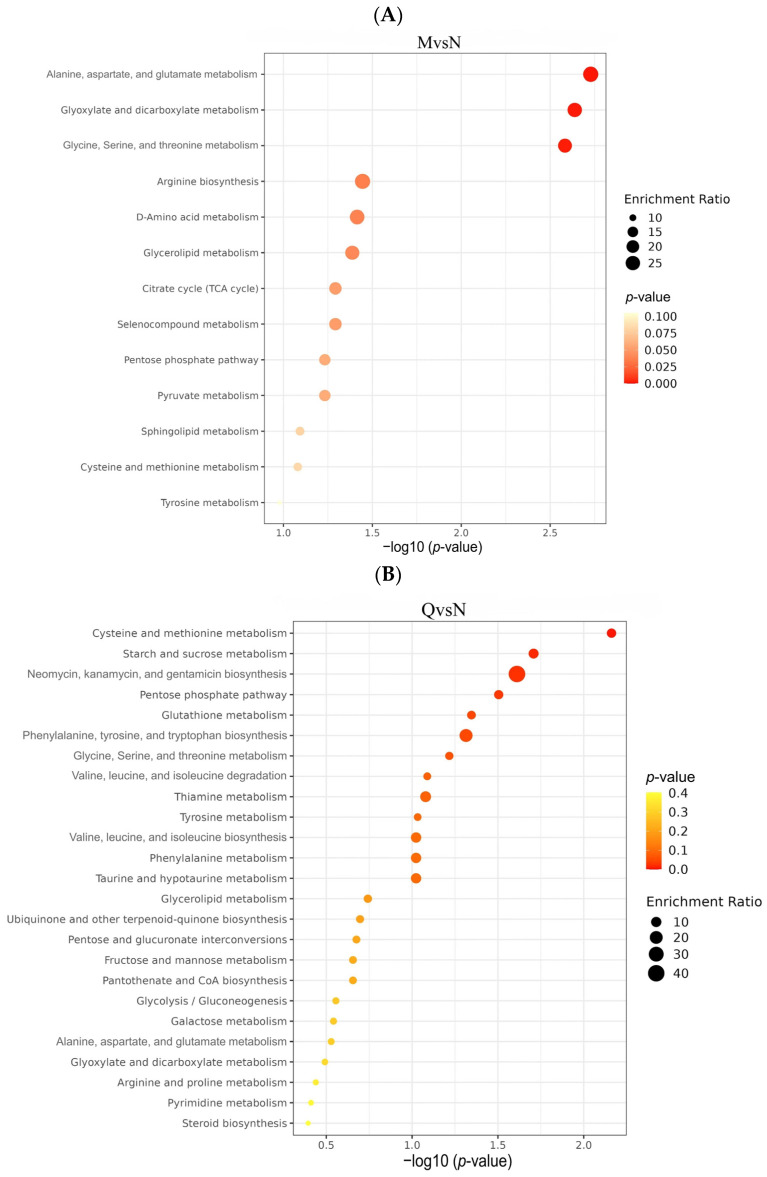
(**A**) Bubble plots of KEGG enrichment analysis of DMs in the NM group. (**B**) Bubble plots of KEGG enrichment analysis of DMs in the QN group. (**C**) Bubble plots of KEGG enrichment analysis of DMs in the QM group. (**D**) The simplified metabolic model based on a heatmap of DMs. The DMs changes were represented by the log2 FC. The color of the border indicated the shift in expression of the corresponding DMs, with red indicating up-regulation and green indicating down-regulation.

**Figure 8 biology-13-00211-f008:**
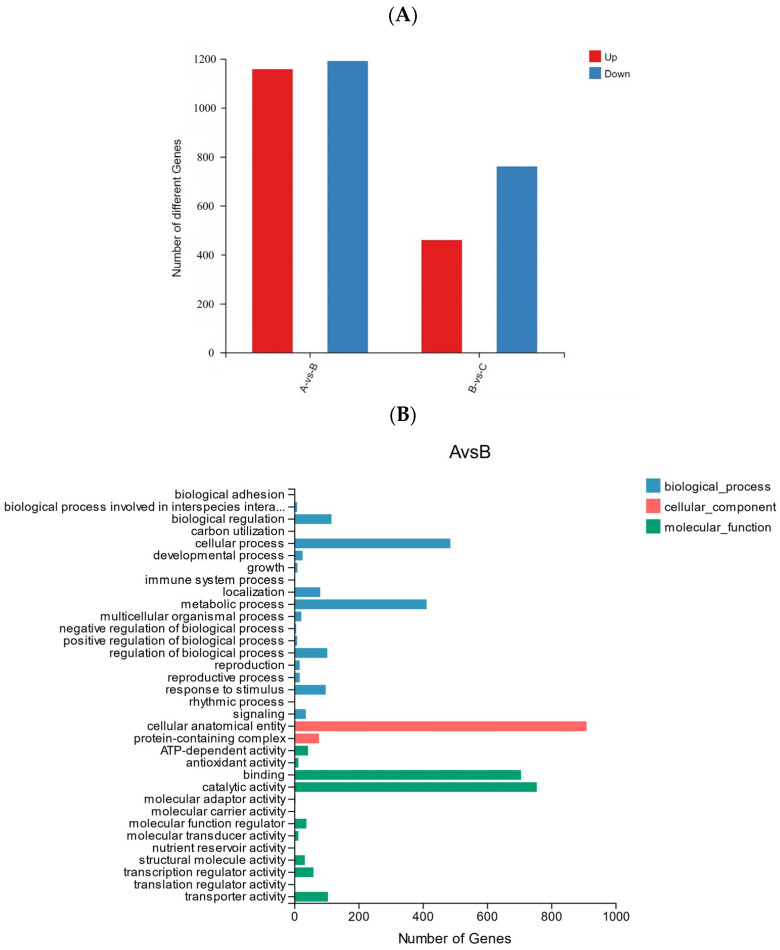
(**A**) Statistics of up-regulated and down-regulated DEGs. (**B**) GO analysis of DEGs in AB comparison group. (**C**) GO analysis of DEGs in BC comparison group.

**Figure 9 biology-13-00211-f009:**
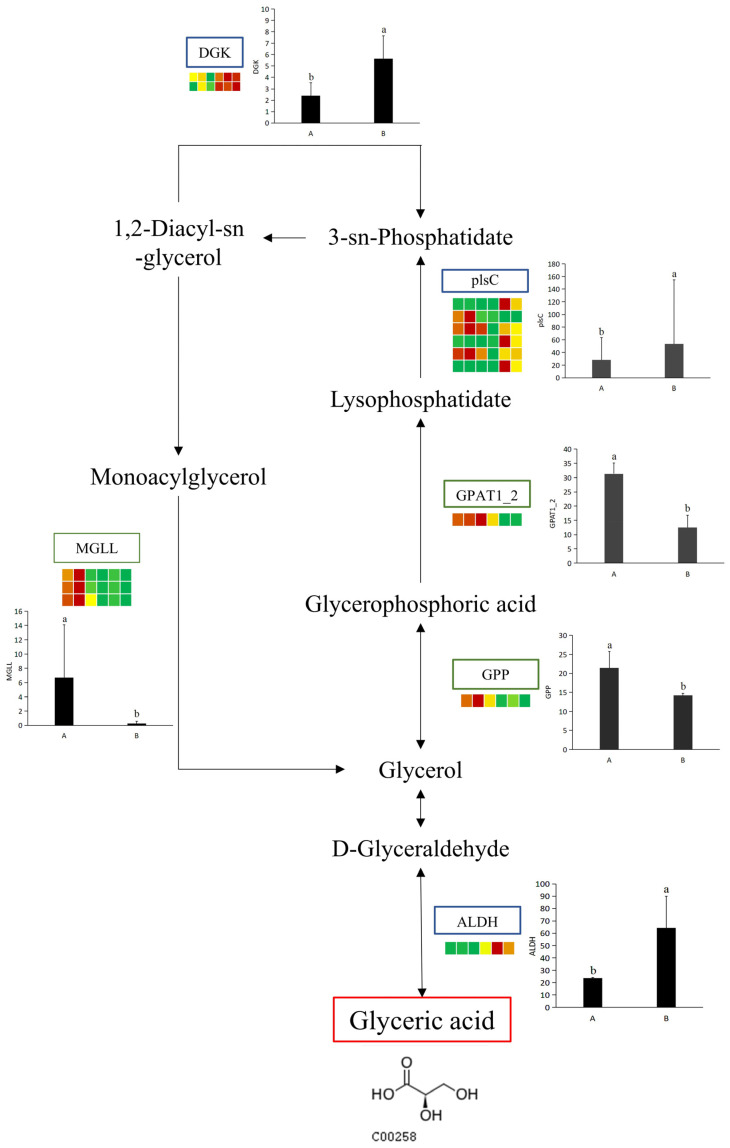
Statistics of up-regulated and down-regulated DEGs. The simplified transcriptome model based on a heatmap of DEGs. DEGs were screen via log2 FPKM. The color of the border means the shift in expression of the corresponding DEGs, with blue indicating up-regulation and green indicating down-regulation. ALDH: aldehyde dehydrogenase; GPP: (DL)-glycerol-3-phosphatase; GPAT1_2: glycerol-3-phosphate O-acyltransferase; plsC: 1-acyl-sn-glycerol-3-phosphate acyltransferase; DGK: diacylglycerol kinase; MGLL: acylglycerol lipase. The letters a and b indicate a statistically significant difference (*p* < 0.05). The length of the bar chart represents the average, and the length of the error bar represents the standard error.

**Figure 10 biology-13-00211-f010:**
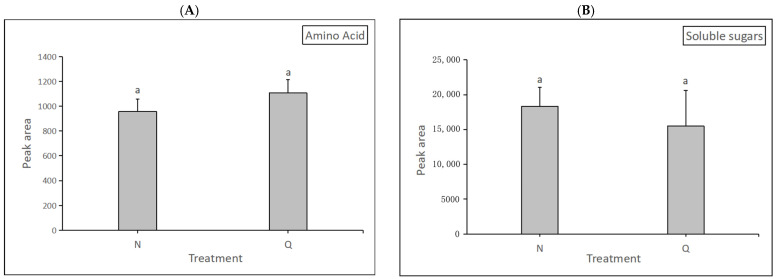
The impact of UV-B on *R. chrysanthum*’s physiological indicators. (**A**) The changes in amino acid content. (**B**) The changes in soluble sugar content. The presence of letter “a” in the figure means that the difference is not significant. The length of the bar chart represents the average, and the length of the error bar represents the standard error.

**Figure 11 biology-13-00211-f011:**
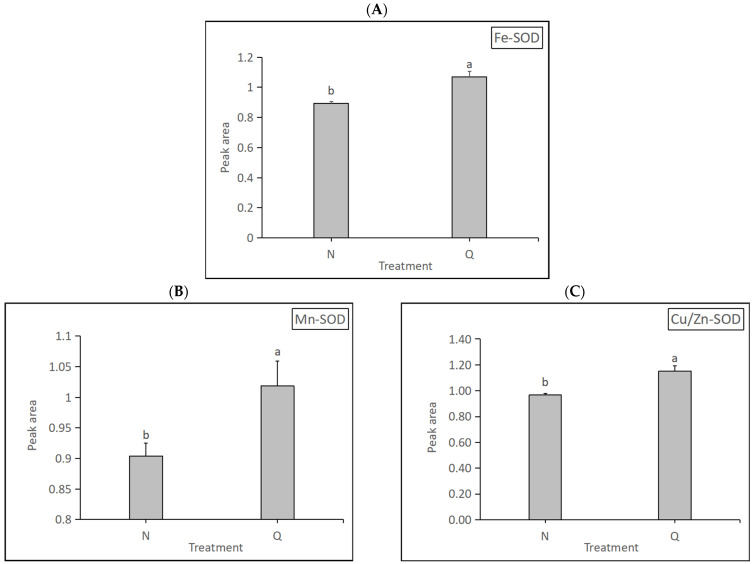
Detection of three kinds of SOD expression changes: (**A**) The changes in amino Fe-SOD expression. (**B**) The changes in Mn-SOD expression. (**C**) The changes in Cu/Zn-SOD expression. The letters a and b indicate a statistically significant difference (*p* < 0.05). The length of the bar chart represents the average, and the length of the error bar represents the standard error.

**Table 1 biology-13-00211-t001:** UV-B-responsive metabolites between different comparison groups were determined using multidimensional statistical analysis.

DMS Type	Class	Name	KEGG ID	*p* Value	FC	Type
UV-B-responsive metabolites	Alcohols	2-Hydroxypyridine	C02502	0.081	1.328	up
	Amino Acid	L-Tyrosine	C00082	0.0019	2.081	up
	Amino Acid	3-Nitrotyrosine	NA	0.0077	0.149	down
	Amino Acid	L-Serine	C00065	0.021	3.151	up
	Amino Acid	3-Aminoisobutanoic acid	C05145	0.022	2.037	up
	Amino Acid	3-Oxoalanine	NA	0.026	1.431	up
	Amino Acid	L-Methionine	C00073	0.031	0.646	down
	Amino Acid	N-Acetyl-L-aspartic acid	C01042	0.042	2.207	up
	Amino Acid	Homocysteine	NA	0.06	1.598	up
	Carbohydrates	Lactulose	C07064	0.034	2.285	up
	Carbohydrates	D-Glucose	C00031	0.059	0.596	down
	Carbohydrates	L-Arabitol	C00532	0.08	2.245	up
	Fatty Acids	Palmitoleic acid	C08362	0.007	0.466	down
	Fatty Acids	Myristoleic acid	C08322	0.02	2.057	up
	Hormone	Normetanephrine	C05589	0.028	2.103	up
	Indoles	Indoleacetic acid	C00954	0.00053	0.288	down
	Lipids	Phytol	C01389	0.0026	0.52	down
	Lipids	Cortisol	C00735	0.036	1.563	up
	Lipids	MG182	NA	0.075	2.398	up
	Nucleotide	Uridine	C00299	0.026	16.377	up
	Nucleotide	Guanine	C00242	0.074	0.394	down
	Vitamin	Alpha-Tocopherol	C02477	0.096	2.113	up
	Organic Acids	Fumaric acid	C00122	0.033	1.57	up
	Organic Acids	Maleic acid	C01384	0.035	1.91	up
	Organic Acids	3-Pyridylacetic acid	NA	0.048	1.963	up
	Organic Acids	3-Hydroxybutyric acid	C01089	0.09	5.289	up
Potential UV-B stress biomarker	Organic Acids	Glyceric acid	C00258	0.0078	1.833	up

**Table 2 biology-13-00211-t002:** The *p* value of less than 0.05 is considered statistically significant for metabolic pathway enrichment.

ID	Pathway Name	Total	Expected	Hits	*p* Value	FDR
map00250	Alanine, aspartate, and glutamate metabolism	28	0.529	5	0.0001	0.00771
map00970	Aminoacyl-tRNA biosynthesis	48	0.906	6	0.0002	0.00771
map00270	Cysteine and methionine metabolism	33	0.623	5	0.0003	0.00771
map00260	Glycine, Serine, and threonine metabolism	33	0.623	3	0.0226	0.459
map00524	Neomycin, kanamycin, and gentamicin biosynthesis	2	0.0378	1	0.0374	0.459
map00350	Tyrosine metabolism	42	0.793	3	0.0424	0.459
map00040	Pentose and glucuronate interconversions	18	0.34	2	0.0437	0.459
map00500	Starch and sucrose metabolism	18	0.34	2	0.0437	0.459

**Table 3 biology-13-00211-t003:** Results of the analysis of *R. chrysanthum* transcriptome sequencing data.

Sample	Total Raw Reads (M)	Total Clean Reads (M)	Total Clean Bases (Gb)	Clean Reads Q20 (%)	Clean Reads Q30 (%)	Clean Reads Ratio (%)
A1	45.44	42.21	6.33	97.75	93.39	92.89
A2	45.44	43.29	6.49	97.58	92.93	95.27
A3	45.44	43.25	6.49	97.52	92.75	95.19
B1	45.44	43.07	6.46	97.67	93.22	94.8
B2	45.44	43.51	6.53	97.44	92.56	95.76
B3	45.44	42.98	6.45	97.67	93.21	94.6
C1	43.69	42.21	6.33	97.66	93.14	96.61
C2	45.44	43.23	6.49	97.73	93.36	95.15
C3	45.44	42.87	6.43	97.65	93.09	94.35

**Table 4 biology-13-00211-t004:** DEGs associated with potential UV-B stress biomarker under UV-B radiation.

Gene Annotation	Gene ID	log2(FC)	A FPKM	B FPKM	Type
ALDH	TRINITY_DN520_c2_g1_i1-C_3	1.39	23.596	64.31	up
GPP	TRINITY_DN27872_c0_g1_i1-A_1	−0.63	21.456	14.18	down
GPAT1_2	TRINITY_DN4514_c1_g1_i1-A_1	−1.38	31.313	12.516	down
plsC	TRINITY_DN1355_c0_g1_i2-C_2	5.15	1.813	64.5	up
	TRINITY_DN16060_c0_g1_i1-C_1	−4.32	44.256	2.126	down
	TRINITY_DN2585_c0_g1_i1-A_1	−1.17	38.766	17.896	down
	TRINITY_DN3143_c1_g5_i1-B_2	4.44	0.136	3.01	up
	TRINITY_DN614_c0_g1_i4-C_1	−0.56	85.326	59.55	down
	TRINITY_DN716_c0_g1_i3-B_2	9.33	0.266	172.753	up
DGK	TRINITY_DN3436_c0_g2_i2-B_3	1.30	1.5	3.713	up
	TRINITY_DN533_c0_g1_i1-C_2	1.16	3.32	7.61	up
MGLL	TRINITY_DN1203_c0_g1_i3-C_1	−6.03	1.806	0.046	down
	TRINITY_DN1835_c0_g1_i4-A_1	−4.69	12.776	0.513	down
	TRINITY_DN2014_c0_g1_i2-C_2	−5.03	5.506	0.183	down

## Data Availability

The data used in this study are available from the corresponding author on submission of a reasonable request.

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
