# Peer review of "Comparative Metabolomics and Transcriptome Studies of Two Forms of Rhododendron chrysanthum Pall. under UV-B Stress"

_biology, 2024, doi:10.3390/biology13040211_

Round 1

Reviewer 1 Report

Comments and Suggestions for Authors

Dear Authors!

The manuscript is devoted to the actual topic of studying plant responses to solar ultraviolet radiation. The manuscript may arouse the interest of readers, but in order to improve its quality, I propose to make some changes to it.

1. Write the full species name of R. chrysanthum in the title of the manuscript and on line 27.

2. Line 30-31. Incorrect links to articles [8-10]. These articles do not mention R. chrysanthum.

3. Lines 38-39. “…Okra made significant daily domestication changes to UV screening with fluctuations in UV-B radiation.” What are these changes?

4. Decipher the abbreviations GO (line 50), SOD (line 91), TCA (Fig. 5), FDR (Table 2)

5. Line 50. What does the phrase “After being transferred to the lab...” mean?

6. Section 2.1 needs correction. The experimental procedure is written very unclearly.

7. Section 2.2. Briefly describe how Soluble sugars, Amino acids and SOD content were determined and provide links to the methods used.

8. Section Results. In general, more detailed comments are needed on the data presented in the figures and tables.

9. In the Discussion section, I recommend referencing the figures and tables from the Results section.

10. It is known that UV-B strongly affects for growth, development, and morphology of plants. Why did the authors not measure the morphological characteristics of both species of R. chrysanthum under UV-B stress?

Comments on the Quality of English Language

Minor editing of English language required

Reviewer 2 Report

Comments and Suggestions for Authors

This manuscript described the metabolomic and transcriptomic analysis for two types of R. chrysanthum; however, several concerns were arisen.

One of the most important is that the authors states that this study was done with two types (Varieties, cultivars or what?) of R. chrysanthum, but in material and methods is clear that experiments were done using the same plant under different conditions, 1) seedlings growing under UV-B, and 2) Seedlings growing under white light, and then BOTH seedlings were exposed to UV-B for 8 h each day for two days. However, in results showed the metabolome analysis of plants in response to UV-B when BOTH treatments were exposed to UV-B. It is important to clarify why authors decided apply UV-B during seedling growing to understand what they want to explain. In metabolome results mention as DMS in response to UV-B, but all samples including controls were exposed to UV-B.

In general, grammar needs to be substantially improved and several spelling errors have to be corrected.

-Nomenclature of control and treatments samples is confusing, are UV-B growing seedlings expose to UV-B (N,Q,) or PAR (B, C)?

-I suggest include a list of differentially expressed metabolites because only fugure 2A is presented. in figure 2E only samples M,N and Q are shown, what are about the others?

-Results need to be more described, authors do not mention specific name of metabolites (only present tables) but no mention their importance nor biological relevance.

-Homogenize UV-B throughout the text, correct the scientific name of plant, correct metabonomics to metabolomics in the title, etc, etc

-A space is lacking in some measures, e.g. line 23, 350nm

Despite the title described a metabolomic and transcriptomic studies, authors only mention a few studies related to metabolomics but no transcriptomics in introduction.

-Lines 48-54 are no proper for introduction and seems an antecedent because authors placed this paragraph before the aims

-Aim of the study are no clear and are not coherent with the title and the presented results.

-Point 2.3 there is no information about how the metabolite were extracted from the samples.

-Point 2.5, methodology for mRNA and sequencing lack of important information regarding how RNA total purification was made, kits and consumables used for construct the library, type of library (read size, single or paired end, etc), instrument for sequencing, etc.

-Citation for methods and softwares used for transcriptomics are lacking.

-What were the criteria used for select a potential biomarkers from DEGs?

-fig 8, bars correspond to standar deviation or error? variation is too big, I am not sure that this treatments will be statistically significant.

-line 306, I do not understand authors conclude that glyceric acid is a UV-B resistance biomarker, no mention in results

Comments on the Quality of English Language

Manuscript need to be edited by a native English speaker. 

Reviewer 3 Report

Comments and Suggestions for Authors

Comments for the manuscript entitled "Comparative Metabonomics and Transcriptome Studies of two types of R. Chrysanthums under UV-B Stress" submitted by Wang Yu et al.

The present study focuses on the complex, molecular-level mechanism of plant response to UV-B stress. To this end, research was carried out on two forms of Rhododendron chrysanthum, namely a domesticated form grown in an artificial climate chamber, and a form grown in the intelligent artificial incubator.

After exposing the seedlings of the two forms of R. chrysanthum to treatments with UV-B  (280-315 nm) and PAR (280-315 nm) were detected the physiological indexes of experimental radiation, metabolites (quantitatively and qualitatively), has been established metabolic pathways enrichment analysis, transcriptomic analysis (establishing the number of differential expressed genes for each experimental group). 

The metabolome and transcriptome research ends with the presentation of changes in physiology for domesticated R. chrysanthum, concluding that:  UV-B significantly increased the content of amino acids; glyceric acid can be a biomarker of resistance to UV-B stress; soluble sugars decreased; the expression levels of the three SOD enzyme types were significantly elevated. All this indicates that domesticated R. chrysanthum is very resistant to UV-B radiation.

My comments are below:

1. In the title it should be replaced "two types of R. Chrysanthums" with two forms of Rhododendron chrysanthum. Also, "Metabonomics" should be replaced with Metabolomics.

2. The abstract should be more consistent, in accordance  with the large amount of data obtained. UVB and UV-B appear. It should uniformity. In abstract, you should clearly specify the essential difference between the two forms of R. chrysanthum. Replace "the two test materials" with the two forms of R. chrysanthum.

3. The purpose (I) of this study is not clearly expressed.

4. Delete the point before the brackets, as you did in lines 47, 321.

5. In the section "2.1. Plant Material and treatment" you did not specify the direct source of origin of the rhododendrons that you transferred to the laboratory. From Mount Changbai? Also, do not specify what kind of organ fragments you have used to obtain explants. What kind of nutritive medium have you used to cultivate explants? Was the composition of the nutritive medium the same for artificial climate chamber and for intelligent artificial incubator? In this section, you followed the development of the two forms of R. chrysanthum. You should emphasize this more.

6. You should specify that abbreviation PAR referring to photosynthetic active radiation.

7. In lines 17, 49, 53, 371, 372 it is R. chrysanthum and not R. Chrysanthum.

8. In lines 290, 308 you wrote "two R. chrysanthum species". We are not talking about two species, but two forms of R. chrysanthum!!!

9. In line 303 it should be UV-B instead of V-B.

10. The conclusions should be reformulated as they do not clearly reflect the results obtained.

11. You should put after "Conclusions" a list of the abbreviations used.

I wish you success in getting this manuscript published after the recommended corrections!

Round 2

Reviewer 2 Report

Comments and Suggestions for Authors

Thank you to authors for attend my suggestions and for clarify most of my doubts, in special those related to experimental design for UV-B exposition; but I still have some comments about fundamental questions that are no answered and clarify, and recommendations.

I feel a better writing is needed in some sections, such as lines 131-135

Line 25, 26 write with letters

Line 30 & 31, eliminate dash

Line 32, acbvity?

line 85, 128, edit crysanthums

line 112 add space after nm

line 114, space after mol

line 102, space after ¼

line 107, it is no clear how much time explants were subjected to UV-B

line 120, “tested materials”

line 144, homogenize GC-TOFMS as previously you used

line 171, authors do not describe how total RNA was obtained, grinded with nitrogen and extracted with TRIZOL or any kit?; moreover, add the brand of kits and consumables.

lines 185-192, are lacking of detailed about library and sequencing, in this pointlines 187-192 are results, not material and methods, I recommend revised published papers related to transcriptomics as a guide.

Line 204, author mention that R. crysanthum has no sequenced genome, but how you aligned clean reads to genome sequences? You have to a de novo assembly transcriptome.

Line 425, results were not significant, I don’t understand you say that soluble sugar decrease.

Many other questions remain to be unanswered.

Comments on the Quality of English Language

Moderate
